# Pragmatic Language Development: Analysis of Mapping Knowledge Domains on How Infants and Children Become Pragmatically Competent

**DOI:** 10.3390/children9091407

**Published:** 2022-09-16

**Authors:** Ahmed Alduais, Issa Al-Qaderi, Hind Alfadda

**Affiliations:** 1Department of Human Sciences (Psychology), University of Verona, 37126 Verona, Italy; 2Institute of English Studies, University of Warsaw, 01-445 Warsaw, Poland; 3Department of Curriculum and Instruction, King Saud University, Riyadh 11451, Saudi Arabia

**Keywords:** pragmatics, pragmatic language development, pragmatic competence, child language development, typical language development, atypical language development, scientometric review

## Abstract

New-borns are capable of recognising and producing sounds as they become phonologically competent. Following this, infants develop a system for connecting these sounds, which helps them become increasingly lexically competent over time. Their knowledge of these words grows as they develop, using words to form phrases, turning them into sentences, and ultimately becoming syntactically competent. By making sense of these linguistic elements, these three competencies are enhanced, and this is how infants become semantically competent. As infants continue to develop linguistic and non-linguistic communication behaviours, this miraculous language development becomes even more complex, enabling them to perfect their linguistic abilities while being pragmatically competent. In this study, a scientometric approach was used to examine past, present, and future trends in pragmatic language development (PLD). A total of 6455 documents were analysed from the Scopus, WOS, and Lens databases between 1950 and 2022. The analysis involved the visualisation and tabulation of eight bibliometric and eight scientometric indicators using CiteSpace 5.8.R3 and VOSviewer 1.6.18 software for data analysis. In this study, we highlight the major patterns and topics directing the research on PLD between 1950 and 2022. The themes and topics included (1) analysing PLD as a social behaviour through the lens of executive functions; (2) studying PLD as a social behaviour based on social understanding; (3) examining PLD as a social behaviour associated with autism spectrum disorder; (4) developing an understanding of PLD in academic settings through the examination of executive functions; (5) identifying pragmatic competence versus communicative competence as a social behaviour; (6) analysing pragmatic language skills in aphasic patients via epistemic stances (i.e., attitudes towards knowledge in interaction); (7) investigating PLD as a behavioural problem in the context of a foreign language; (8) assessing PLD as a behavioural problem in individuals with autism spectrum disorder; (9) assessing PLD in persons with traumatic brain injury and closed head injury as a behavioural problem; (10) identifying the role of the right hemisphere in executive functions as a cognitive substrate; (11) assessing the impact of pragmatic failure in speech acts on pragmatic competence; and (12) investigating the patterns of PLD among learning-disabled children.

## 1. Introduction

### 1.1. The Rise of Pragmatic Language Development 

The study of pragmatic language development (PLD) seems to have first emerged in the mid-1970s. Halliday [1] was one of the pioneers, and began to analyse early child communication within the framework of the Speech Act Theory [2,3,4]. In around 1980, several co-authored books were published that addressed all or part of this developmental stage [5,6,7]. Since then, important progress has been made in several directions. The book by Ninio and Snow [8] is noteworthy for its substantial contribution to the entire field of early PLD. PLD is a component of the whole language system, which is viewed as a tool that children and adults use to explore the social world; create, develop, and maintain social relationships; and engage in culturally significant activities with others [9]. As social animals, it is impossible for speakers to survive without language, which allows them to communicate their own experiences, thoughts, beliefs, and ideas. Humans are distinguished and superior to all other creatures in the universe due to their ability to communicate through language [10].

PLD has drawn the attention of many scholars for several reasons. First, children seem to master lots of pragmatic functions at a time when their vocabulary and syntax are still limited [1,2,11,12]. Ingram [11], for example, presented evidence for the rapid expansion of the range of pragmatic functions during the one-word and very early two-word phases. However, the evidence for this suggestion was based on several different studies rather than on a comprehensive assessment of the same children. Halliday [1] and Keller-Cohen et al. [1] claimed, on the basis of intensive studies with one and two children, respectively, that a relatively universal sequence of the emergence of functions can be observed. Therefore, development during this period may manifest itself in functions rather than in lexical or syntactic forms [13]. Second, there is empirical evidence that pragmatic development is a statistically independent dimension of development. Snyder [14] compared language-delayed children with normal children matched for mean utterance length in terms of their ability to produce declarative and imperative functions under structured elicitation conditions and found that the language-delayed children were more delayed in pragmatic than in syntactic terms. An even clearer example of this independence was provided by Blank et al.’s [15] case study of a boy who, despite relatively well-developed syntactic and semantic systems, showed an almost complete deficit in the ability to use language socially and appropriately. Third, it is often assumed that pragmatic development is the aspect of language most closely related to cognitive development [16]. In normal children, Piaget’s period of one-word utterances (very roughly estimated to span 12 to 20 months of age) coincides with major cognitive changes. Pragmatic measures, if feasible, could be much more fruitful than measures of vocabulary and syntax for studying the relationship between language and cognitive development. All three of these issues could be studied more effectively if appropriate measures of PLD were available and could be used, for example, for correlational studies [16].

### 1.2. Pragmatic Language Development: Infancy

Children use language to express their needs and wants, negotiate disagreements, participate in games, and engage in other communicative interactions with peers and adults. Basic pragmatic skills develop at a fairly early age but are refined and developed during preadolescence and adolescence, so that over time the child is able to participate in more social activities and become a full member of culture and society [17]. Children’s pragmatic competence—their ability to use language effectively—is developed and refined through participation in family, peer, and school interactions that serve as a means and motivation for efficient and strategic language use. By examining both universal and culturally specific features of children’s interactional skills, studies of pragmatic development aim to describe and detail how children learn to use language as an effective tool for social interaction [18].

One of the basic rights of children is to have a family that takes care of their basic needs and provides them with affection and social support [19]. Pragmatics is a process of social development in which the knowledge necessary for successful interaction with others is acquired through language. This ability is not acquired suddenly; the child must develop skills such as the ability to process information from different sources, appropriate linguistic development, and the ability to respond to social demands [19,20].

When children learn a language, they need to learn more than phonology, semantics, and syntax [21]. A proficient language user knows how to use language appropriately and strategically in social situations. Children need to learn pragmatic skills, also known as communicative competence. They need to learn how to use language in interactions with peers, families, teachers, and others. Children need to learn how to ask questions, make requests, give orders, agree or disagree, apologise, refuse, joke, praise, and tell stories. They need to learn routines such as “trick or treat” and “happy birthday”; polite expressions such as “please” and “thank you”, “hello” and “goodbye”, and “excuse me”; and how to address others. They need to learn how to start, maintain, and end conversations; when to speak or remain silent; how to take turns; how to give and respond effectively to feedback; and how to stay on topic. They need to know and use the right volume and tone of voice. They need to learn how the meaning of terms such as “I” and “you” or “here” and “there” changes depending on who is speaking and who is listening. They need to learn what style of speech to use, when to use jargon, and when and whether to talk about certain topics [21].

Between 9 and 18 months of age, the first milestones of PLD are reached, such as understanding the communicative intent of the speaker and the onset of joint attentional behaviour [22,23,24,25,26,27]. Toward the end of the second year of life, infants become more adept at identifying what others want, intend, and perceive. Meltzoff [28], using the behavioural imitation paradigm, found that 18-month-old infants will imitate an adult’s action even if it fails, but will not repeat that action when performed by a mechanical device. This study and subsequent studies with some variations in experimental conditions [29,30,31,32,33,34,35,36,37] showed that 18-month-old infants are able to recognise that people, but not animate things, have intentions and are able to infer the intentions of others. In addition, most 18-month-old children have mastered the ability to appropriately match their responses to specific stable characteristics of their conversational partner [38] and to the type of activity they are engaged in with their partner [39,40].

At age 2, children appear to be able to answer simple questions whose responses require only a single element, whereas, at age 3, children are able to make comments and give responses that include both the predicate and the direct object [41]. In addition, an increasing ability to use complex contextual information is observed, which is reflected in 3-year-old children’s ability to answer and formulate questions and comments [42,43]. “Where” and “what” questions are easier for children to answer than “why” questions; however, it is not uncommon for children to use a particular question form in one context but not in another. It is possible that this discrepancy is not due to difficulties with the linguistic form, but rather situational and pragmatic cues that may play a larger role in children’s performance [44].

As toddlers and preschoolers progress, pragmatic skills continue to develop, and children gain more advanced conversational skills and become socially competent speakers (O’Neill, 2007) [45]. Four-year-olds are able to adapt their own language to take into account the age, gender, and status of the listener. Thus, preschoolers start to formulate less polite and more direct requests when interacting with higher-status speakers (e.g., parents and teachers) and more polite and indirect requests when interacting with lower-status speakers (e.g., friends). They develop very detailed expectations about what terms can be used to talk about a particular situation. Specifically, in terms of developing reference choice, preschoolers prefer optimally informative referential terms to both under-informative and over-informative equivalents [46,47,48].

### 1.3. Pragmatic Language Development: Adolescence

As children enter adolescence, their growing social world both enables and pressures them to develop more sophisticated pragmatic skills. Experience with a greater variety of teachers and peers, exposure to more forms of language through reading and school, and participation in extracurricular activities motivate adolescents to take the perspective of others and use language strategically. In addition, pragmatic behaviour reflects normal progress in identity development and increasing autonomy from parents. The social contexts in which adolescents engage in pragmatic behaviour also include technologies such as mobiles and the internet [21].

During adolescence, language and pragmatics play a particularly important role in identity formation and marking [21]. Family conversations and narratives connect adolescents to their past and their culture. The appropriate use of current slang expressions and gestures typical of the peer group is crucial. These behaviours show solidarity with members of the groups to which the adolescents belong and clarify the adolescents’ distinction from other groups and from younger children and adults. Knowing the current terms for the groups themselves and knowing how to tease and argue become more important during the teen years. Shifting between different registers and varieties of language allows adolescents to associate themselves with particular age, gender, social class, racial, and ethnic groups. Even gossip, insults, and verbal aggression in relationships serve to forge alliances and cross social boundaries, discover norms among peers, and explore identity [21].

### 1.4. The Scope of Pragmatic Language Development

A search of the pertinent literature yielded a number of varied and general definitions of pragmatic language. Gallagher defines pragmatic language as “linguistic elements and contextual elements as forming a contextual whole” [7] (p. 2). A simpler definition is by Bates [16], who defines pragmatic language as “rules governing the use of language in context” (p. 420). Other researchers do not offer definitions that provide insight into the parameters of pragmatic language, although they acknowledge the complexity of pragmatic language skills [49].

The term pragmatic language refers to the use of language in the context of communication [50,51]. This broad definition encompasses many different pragmatic language skills and abilities, including the ability to sustain a conversation, provide relevant responses, follow politeness norms, write coherent reports, and understand non-literal language such as jokes, sarcasm, and irony [51]. Pragmatic language is defined as a complex and uniquely human skill that is embedded in children’s daily experiences. It enables us-children and adults alike to form relationships with one another, share experiences, and communicate our perspectives and attitudes about those experiences [52,53,54]. Pragmatic language is a multifaceted construct that involves many functional capacities, including cognitive, linguistic, and theory of mind [55,56,57,58].

Pragmatic language refers to the appropriate and effective use of language in interpersonal contexts and is central to children’s ability to perform well at home, at school, and with peers [59,60,61]. It can be distinguished from the structural aspects of language, which have traditionally been viewed as relatively independent of context: phonology, syntax, and semantics. Difficulties in the pragmatic language domain are manifested in a variety of behaviours, such as talking too much; taking turns poorly in conversation; the inability to adapt a message to a listener’s needs; the inability to respond to verbal cues from others; the overuse of stereotypical phrases; and difficulty understanding sarcasm, jokes, and metaphors [58,59,60].

Pragmatic language has been studied in the disciplines of anthropology, sociology, psychology, and linguistics [5]. In the last two decades, cognitive–behavioural psychology, linguistics, and social cognitive psychology have had a significant impact on the development of pragmatic language theories. Each of these traditions offers a different perspective on the boundaries of pragmatic language and the appropriate unit of analysis.

PLD refers to children’s linguistic and non-linguistic communication skills, which include various influences such as socialisation by caregivers, parents, siblings, teachers, and peers; cognition; knowledge; and effort (Bryant, 2018, as cited in [62]). Pragmatic language is defined as the use of appropriate communication in social contexts; in other words, knowing what to say, how to say it, and when to say it [63]. Pragmatic skills enable children to produce and understand words and sentences in ways appropriate to the conversational context [64]. According to the American Speech–Language Association (ASHA), social communication consists of three major communication skills, namely using language, changing language, and following rules [65], as is shown in Table 1.

### 1.5. Scientific Contributions for Pragmatics

Table 2 shows the 10 most important source journals for PLD research along with a brief description of the scope of each source journal. The title of the source journal, the country in which the journal was published, the name of the publisher, the starting date, the number of volumes published to date, and web addresses are also outlined.

### 1.6. Purpose of the Present Study

In recent years, pragmatics and PLD have experienced a significant increase in research [66]. Whether infants and children can undergo typical or atypical PLD has been researched internationally in both school and clinical settings [67]. Recent research has examined how children develop pragmatic language skills in school and clinical settings in light of the different theories and assessment methods for PLD. This review is significant in that it provides an overview of how pragmatic language skills are acquired from infancy to adulthood [68]. In light of this, the present study examines PLD from a bibliometric and scientometric perspective. We attribute the significance of this study to the presentation of historical evidence regarding PLD, the analysis of existing evidence concerning PLD, and the prediction of future trends related to PLD research. We raised the following questions: (1) What is the size of knowledge production in PLD measured by year, region, higher education institution, journal, publisher, and author? (2) What are the most cited documents in PLD? (3) Who are the most influential authors in PLD? (4) Which topics and themes are most frequently explored in PLD? (5) Which research trends are emerging in PLD?

## 2. Methods

### 2.1. Research Methods

Scientometrics is defined as “the study of artifacts; one examines not science and scholarship but the products of those activities” [69] (p. 491). It is usually the objective of researchers in this field to analyse “the quantitative aspects of the production, dissemination and use of scientific information with the aim of achieving a better understanding of the mechanisms of scientific research as a social activity” [70] (p. 6). Researchers debate whether scientometric studies can be used to assess the quality of published research. In a previous study, it was identified that “the task of determining quality papers is especially difficult in BIS [bibliometrics, informetrics and scientometrics] due to the very heterogeneous origin of the researchers” [71] (p. 390). There is, however, a purpose in these studies, which remains to provide “reveal characteristics of scientometric phenomena and processes in scientific research for more efficient management of science” [72] (p. 1).

Several scientometric indicators have been developed to guide researchers in the conduct of such studies. These indicators can be categorised as either referring to elements (e.g., publication, citation and reference, and potential) or type indicators (e.g., quantitative, impact) [72]. “Mapping knowledge domains” also merits our attention, and it refers to the process by which we can produce “an image that shows the development process and the structural relationship of scientific knowledge”—using maps that are “useful tools for tracking the frontiers of science and technology, facilitating knowledge management, and assisting scientific and technological decision-making” [73] (p. 6201). In recent research, it has been argued that this method should be applied to all fields of research, not just medical, health, and pure sciences [74]. In the context of this study, the field of PLD is explored as a sub-field of pragmatics, one that integrates other fields, including linguistics, psychology, and education.

### 2.2. Measures

As mentioned above, both bibliometric and scientometric studies are considered tools to guide the assessment of the knowledge produced in a particular field/concept (e.g., PLD) by evaluating the production of documents. Bibliometric indicators are usually provided in knowledge databases (e.g., Scopus, WOS, and Lens) [75,76,77,78]. The scientometric indicators are generally provided by scientometric software, which is part of the scientometric analysis. During the course of this study, for example, we used CiteSpace 5.8.R3 [79] and VOSviewer 1.6.18 [80]. A summary of the bibliometric and scientometric indicators we used in this study can be found in Table 3.

### 2.3. Data Collection and Sample

Three databases—Scopus, WOS, and Lens—were used to retrieve the data. These databases were included for a number of reasons. To begin, both Scopus and WOS contain sources that have been evaluated for inclusion based on their quality in addition to being knowledge databases [75,76,77]. Second, the Lens database is regarded as more exhaustive than the first two databases, since it provides data that are unavailable from the other two [78].

A search of the data was conducted on Friday, 25 March 2022. No language limitations were included, provided that the title, abstract, and keywords were in English. Considering the fact that there were few results in other languages, we verified this manually. Articles, review articles, book chapters, and books, including early-access publications, were the only types of documents we considered. The search strings are shown in Table 4, along with other specifications used in the three databases.

To measure the development and size of the research produced in this area, we examined the use of the concept “pragmatic language development” and any synonyms. The keywords we used when searching for works did not include specific works that were restricted to a particular age group, a specific type of learner, a certain language, or any other limitation. According to our preliminary search on Google, as well as our previous experience in the field, we determined that it was best to use the search strings above for identifying knowledge related to PLD. Pragmatic development is defined as “a heterogeneous field on a range of topics associated with the study of how young children develop the skills to use language effectively and appropriately in social interaction” [84] (p. 300). Pragmatic competence is defined as “is a system of knowledge that is neither a system of ‘knowing how’ nor a system of ‘knowing that’… the system of knowledge that governs use of language” [85] (p. 67).

### 2.4. Data Analysis

Several steps were taken before and during the process of analysing the data. To begin with, the data for the bibliometric analysis were exported from Scopus in three different formats: Excel sheets for the bibliometric analysis, CiteSpace RIS files, and VOSviewer CSV files. RIS files from CiteSpace were converted into WOS files in order to fulfil the requirements imposed by CiteSpace. Furthermore, WOS data were retrieved in two formats: text documents converted into Excel sheets for bibliometric analysis, and plain text documents for CiteSpace and VOSviewer. The Lens data were retrieved in two formats: CSV for bibliometric analysis and full record CSV for VOSviewer.

As a preliminary step, duplicate documents were removed from CiteSpace and Mendeley before starting the analysis in CiteSpace. A bibliometric analysis was carried out using Microsoft Excel for the purpose of this study. The tables were generated in Excel and then converted into figures in order to create the citation reports.

All the scientometric settings were set to default in both software packages in order to perform the analysis. We created separate visualisations for each of the three databases, such as network visualisations, overlay visualisations, and density visualisations. For Scoups and WOS, the analysis was carried out three times each: co-occurrence analysis by the keyword of the author, co-citation analysis by the source, and co-citation analysis by the cited author. Four analyses were conducted for Lens: the co-occurrence analysis by author keywords, citation analysis by author, citation analysis by source, and citation analysis by document. For CiteSpace, the analysis was performed three times for Scopus and WOS: co-citation by document (references), co-citation by cited author, and occurrence (keywords). We prepared narrative summaries, cluster summaries, visual maps, and burst tables based on the data we gathered.

## 3. Results

### 3.1. Result Overview

The results of the study can be divided into two sections. In the first section, bibliometric indicators for PLD are presented. These indicators were derived from a number of databases including Scopus, WOS, and Lens and were based on retrieved data. Bibliometric indicators included, for example, publications by year and top 10 countries, universities, journals, publishers, subjects/research areas, and authors. In the second section, we discuss the scientometric indicators for PLD. A combination of CiteSpace and VOSviewer software was used to analyse these indicators. These indicators included citations, co-citations, and co-occurrences.

The first section presents the bibliometric indicators for the study of PLD. First, we introduced the total number of included studies, their type, time span, and knowledge production size by year. Second, we presented the knowledge production size of PLD by country and university and/or research centre. Third, we presented the top journals and publishers disseminating research in PLD. Fourth, we presented PLD knowledge production classified by research area, keywords, and co-occurrence. The bibliometric indicators section is concluded by presenting the top authors contributing to PLD research. The second section presents the scientometric indicators, starting with the strongest citation bursts for keywords in PLD. This is followed by visualisations for co-occurrence, (co)-citation by author, document, and journal. This section is concluded by a tabulation of the most cited documents, top clusters, citation counts, detected bursts, central authors, and sigma metrics for research in PLD.

### 3.2. Bibliometric Indicators for the Study of Pragmatic Language Development

#### Overview of PLD Studies from Scopus, Web of Science, and Lens

A total of 1470 PLD papers were retrieved from Scopus, 1063 from WOS, and 3922 from Lens for the purpose of analysis. In each of the three databases, the data period was 1973–2022, 1985–2022, and 1950–2022, respectively. Scopus contained 1201 articles, 100 review articles, 149 book chapters, and 20 books. The WOS documents included 1022 articles, 38 review articles, 65 book chapters, and 24 early-access articles. In Lens, there were 2191 articles, 104 unknown articles, 255 book chapters, 135 books, 178 dissertations, and 28 preprints. In addition to English, other languages were included, such as Spanish, French, German, Korean, Russian, and Italian. Due to the fact that the analysis was based on the title, keywords, abstract, and references, all the papers included this information in English. These papers were included in order to avoid bias towards data published only in English language publications.

Figure 1A–C show the length of production by year for the three databases. It can be seen that there has been a considerable rise in the production of knowledge in PLD, with the peak of knowledge production occurring in 2021 in Scopus with 159 publications, 2021 in WOS with 123 publications, and 2020 in Lens with 286 publications. The range of publications per year was 1–159 in Scopus, 1–123 in WOS, and 1–286 in Lens. In all databases, the lowest number of publications occurred in the previous year. Further, of 6455 documents in PLD, there were 5928 documents published between 2000 and 2022. It is thus true that over the last two decades, there has been an increase in the production of knowledge related to PLD.

### 3.3. Production of PLD Research by Country and University

Figure 2A–C show the top 10 countries for producing knowledge related to PLD. There is no doubt that the US achieved the highest ranking among the three databases, with a number of publications that appeared to be significantly higher than those of the other countries. The second and third positions in all three databases were exchanged between the UK and China. Aside from North America, Europe, and Australia, the list also included countries in Asia (e.g., China, Japan, Indonesia, and Iran) as well as other countries around the world. It should be mentioned that the selection of these leading countries was based on the ranking created by each database, whereby each author is presumably considered regardless of their order or role, and the 10 countries with the greatest number of authors are listed as the top 10. This was also applicable for universities.

Figure 3A–C present the top 10 universities and/or research centres for producing knowledge related to PLD. The list of the top institutions varied according to the database. While the first university in Scopus is located in the USA, the first in WOS is located in the UK, and the first in Lens is located in Iran. There are two notable institutions from Iran that are located outside the Northern Hemisphere: Islam Azad University and Allameh Tabataba’i University. The rest of the universities are located in North America, Europe, and Australia.

### 3.4. Production of PLD Research by Journal and Publisher

Figure 4A–D demonstrate the top 10 journals publishing research in PLD. While we found journals that include the word “pragmatics” (for example, *Journal of Pragmatics*, *Intercultural Pragmatics*, and *Historical Pragmatics*), there are also journals in the field of linguistics that have this word as part of their title. An extended list of journals based on their publishers is shown in Figure 4D.

Figure 5A,B show the lists of the top 10 publishers for knowledge in PLD. Due to the fact that Scopus does not include publisher information, these lists are limited to WOS and Lens databases. Although “Elsevier” and “Wiley” achieved the highest ranking in both of the databases, the rankings for the rest of the publishers varied between the databases. As an example, “Cambridge University Press” was ranked 7th in WOS, while it was ranked 9th in Lens.

### 3.5. Production of PLD by Research Area, Keywords, and Co-Occurrence

While PLD is extensively studied as a branch of pragmatics, it is also integrated with a vast array of other fields, as seen in Figure 6A–C. As depicted in Figure 6A, the four most widely published subject areas involving PLD are the social sciences, arts and humanities, psychology, and medicine. Figure 6B reveals that the four most prominent research fields associated with PLD are linguistics, educational research, psychology, and rehabilitation. The findings are further supported by Figure 6C, which identifies psychology, linguistics, pragmatic competency, and pragmatics as the top four study areas. There are other topics that could be categorised as PLD-related, and Lens displayed some of the more specific ones, including pragmatic competence, pragmatics, competence, and language use.

### 3.6. Production of PLD Knowledge by Authors

PLD is unquestionably a broad field that does not have a clearly defined number of contributors, as even a single article is considered to be a contribution to the field of PLD. Despite this, we tried to show the authors whose work produced the most knowledge related to PLD, as shown in Figure 7A–C. It can be seen that Taguchi [86] was the first-ranked author in both Scopus and Lens, while Bosco [87] was the first-ranked author in WOS.

### 3.7. Scientometric Indicators for the Study of PLD

#### Overview of PLD Studies from Scopus, Web of Science, and Lens

This section provides a scientometric analysis of the data retrieved from the Scopus, Web of Science, and Lens databases. Specifically, it highlights the impact of a number of concepts, references, and emerging trends on the field of neurolinguistics as well as the impact of certain authors.

For the purpose of this study, we first identified the top keywords with the strongest citation bursts using CiteSpace for data obtained from Scopus (Figure 8A,B). The green line indicates the period during which all the research was conducted. An indication of the beginning and the end of the burst period can be seen by the red line. The word with the strongest citation burst in Scopus was (psychology = 12.5) between 2014 and 2020, and in WOS it was (interlanguage = 7.05) between 2009 and 2015. The citation burst may differ depending on the database. For instance, the keywords in Scopus were more related to the study of pragmatics in clinical settings (e.g., physiology, language disorders, persons with hearing impairments). On the other hand, the keywords in WOS were more related to learning settings (e.g., children, adult, instruction).

A network visualisation of these clusters and authors is also used to illustrate these features (Figure 9A–D). It can be seen in Figure 9A that topics such as pragmatic competence, cognitive pragmatics, hearing impairment in children, and pragmatic performance, among others, are some of the most frequently explored topics in the area of pragmatic learning and development. Figure 9B shows more specific concepts, such as pragmatic development, executive functions, and young children. In Figure 9C,D, the most cited authors and topics are displayed. Among these topics are executive functions, social understanding, and autism spectrum disorder (see Figure 9C). Other topics such as foreign language, speech acts, and EFL learners are included in the WOS database (see Figure 9D). It is easier to read the data in these figures if you think of each cluster in terms of the text that′s next to it. The topic in the mentioned cluster is searched for more frequently the more intense the text is. 

The co-occurrence of keywords is another important factor. Through the use of VOSviewer, we were able to generate three visual network maps depicting the occurrence of the keywords most commonly used in PLD across the three databases (Figure 10A–C). The colours represent different directions that can be studied in relation to PLD. Yellow represents pragmatic competence, blue represents theory of mind, green represents clinical and neuropragmatics, and red represents speech acts (See Figure 10A). Depending on the database, these colours may change. As shown in Figure 10B, green represents pragmatic competence, purple represents pragmatics, and red represents pragmatic development. Figure 10C shows keywords related to autism and PLD in pink.

We generated three visual network maps for co-citations and citations by authors using VOSviewer (Figure 11A–C). Each colour represents a co-citation or citation network. The circle sizes increase with the number of co-citations or citations the author has. According to Figure 11A, Kasper [88], Bosco [87], Bishop [89], and Cummings [85] are the most co-cited authors. Some of the same authors appeared in the Scopus database with others, such as Siegal [90] (see Figure 11B). Using the Lens database, Figure 11C shows that Taguchi [91], Becker [21], etc., were the most cited authors.

We generated three visual network maps for co-citations and citations by source using VOSviewer (Figure 12A–C). Each colour represents a co-citation or citation network. The larger the circle, the more cited or co-cited the source is. In Figure 12A, *Journal of Pragmatics*, *Journal of Autism*, and *Applied Linguistics* appear to be the most co-cited sources. The sources in Figure 12B are similar to those in Figure 12A, but with more significant journals (e.g., *Journal of Child Language*). The citation network for journals is shown in Figure 12C, including *Frontiers in Psychology* and *East Asian Pragmatics*.

The top 10 cited works were derived from the bibliometric data provided in Scopus, WOS, and Lens. This group of documents was merged, and duplicates were removed as shown in Table 5. While Scopus and WOS identified articles as the most cited works, Lens identified books as the most cited works. These top cited papers could represent the leading research in PLD.

### 3.8. Impact of Research on PLD by Clusters, Citation Counts, Citation Bursts, Centrality, and Sigma

#### 3.8.1. Clusters

The network was divided into 15 co-citation clusters in the Scopus data (see Table 6 for the full list of clusters.). The largest six clusters are summarised as follows. The largest cluster (#0) has 236 members and a silhouette value of 0.737. It is labelled as executive function by LLR, pragmatic development by LSI, and social behaviour (3.89) by MI. The most relevant citer to the cluster is Ren [111] “L2 pragmatic development in study abroad contexts”.

The network was divided into 16 co-citation clusters in the WOS data (See Table 6 for the full list of clusters.). The largest six clusters are summarised as follows. The largest cluster (#0) has 214 members and a silhouette value of 0.737. It is labelled as foreign language by LLR, pragmatic development by LSI, and behavioural problem (3.33) by MI. The most relevant citer to the cluster is Bella [112] “Length of residence and intensity of interaction: modification in Greek L2 requests”.

#### 3.8.2. Citation Counts

In Scopus, the top-ranked item by citation counts was Kasper [88] in cluster #0, with citation counts of 380. The second-ranked item was Bardovi-Harlig [113] in cluster #0, with citation counts of 300. In WOS, the top-ranked item by citation counts was Kasper [114] in cluster #0, with citation counts of 258. The second-ranked item was Bardovi-Harlig [115] in cluster #0, with citation counts of 187. The remaining top citation counts for PLD can be found in Table 7.

#### 3.8.3. Bursts

In Scopus, the top-ranked item by bursts was Taguchi [91] in cluster #0, with bursts of 12.35. The second-ranked item was Faerch [128] in cluster #0, with bursts of 10.16. In WOS, the top-ranked item by bursts was Prutting [129] in cluster #1, with bursts of 12.76. The second-ranked item was Mcdonald [130] in cluster #2, with bursts of 8.59. See Table 8 and Figure 13A–D for the remaining top bursts detected in PLD.

#### 3.8.4. Centrality

In Scopus, the top-ranked item by centrality was Bates [137] in cluster #1, with a centrality of 148. The second-ranked item was Brown [116] in cluster #0, with a centrality of 120. In WOS, the top-ranked item by centrality was Kasper [114] in cluster #0, with a centrality of 97. The second-ranked item was Bardovi-Harlig [115] in cluster #0, with a centrality of 96. The remaining central authors in PLD are listed in Table 9.

#### 3.8.5. Sigma

In Scopus, the top-ranked item by sigma was Bates E (1977) in cluster #1, with a sigma of 0.00. The second-ranked item was Brown P (1987) in cluster #0, with a sigma of 0.00. The top-ranked item by sigma was KASPER G (1998) in cluster #0, with a sigma of 0.00. The second-ranked item was Bardovi-Harlig K (2001) in cluster #0, with a sigma of 0.00. See Table 10 for the sigma metrics of the remaining authors in PLD.

## 4. Discussion

The focus of the present study was to investigate PLD as a subfield of pragmatics that is integrated with linguistics, psychology, education, etc. This objective was achieved by analysing the growth of PLD and presenting bibliometric and scientometric data. In the first section, bibliometric indicators such as publications by year and the top 10 regions, universities, journals, publishers, subject/research areas, and authors were provided. In the second section, scientometric indicators, such as citation, co-citation, and co-occurrence, were introduced.

Seven essential points regarding bibliometric indicators were identified: (1) Knowledge output in PLD increased over the past two decades, reaching a peak in 2021 for Scopus and WOS and in 2020 for Lens. (2) The United States ranked first in all three databases, while the United Kingdom and China alternated in second and third place. (3) US universities dominate only Scopus, whereas UK universities and Iranian universities have supplanted them in WOS and Lens, respectively. (4) The journals containing the word ‘pragmatics’ (such as *Journal of Pragmatics*, *Intercultural Pragmatics*, and *Historical Pragmatics*) appear to be the most pertinent. (5) Elsevier and Wiley are the leading publishers for WOS and Lens. (6) PLD-related topic areas include social sciences, arts and humanities, psychology, and medicine. (7) Taguchi [91] and Bosco [166] are the top contributors in the field.

When combined with scientometric indicators, these bibliometric findings have at least five implications. First, identifying the most frequently used keywords assists researchers in following the most commonly researched topics and themes in PLD. For instance, in this study, the most cited keywords included psychology [90], speech act [67], language disability [167], physiology [168], and language disorder [45]. Another list included interlanguage [169], closed head injury [170], pragmatic language skills [171], study abroad [172], and instruction [173]. As evidenced by the keywords listed above, PLD is analysed in terms of three patterns. First, researchers present evidence for PLD in clinical settings and clinical populations by considering various types of disorders. Second, other researchers investigate PLD in individuals with head injuries. Thirdly, researchers who study second language acquisition investigate PLD in an effort to determine how PLD develops in first language vs. second language acquisition/learning.

The second implication relates to data grouping. We mentioned earlier that 6455 documents related to PLD were included. These documents contained the keywords that we used in the search strings. Therefore, in this study, we were able to classify all of these datasets into distinct patterns of interrelation in order to generate clusters that guide PLD research. These clusters included executive function [51], social understanding [167], autism spectrum disorder [174], foreign language [169], traumatic brain injury [170], and pragmatic competence [64]. Overall, these clusters contain the following 12 patterns:(1)Analysing PLD as a social behaviour through the lens of executive functions;(2)Studying PLD as a social behaviour based on social understanding;(3)Examining PLD as a social behaviour associated with autism spectrum disorder;(4)Developing an understanding of PLD in academic settings through the examination of executive functions;(5)Identifying pragmatic competence versus communicative competence as a social behaviour;(6)Analysing pragmatic language skills in aphasic patients via epistemic stances (i.e., attitudes towards knowledge in interaction);(7)Investigating PLD as a behavioural problem in the context of a foreign language;(8)Assessing PLD as a behavioural problem in individuals with autism spectrum disorder;(9)Assessing PLD in persons with traumatic brain injury and closed head injury as a behavioural problem;(10)Identifying the role of the right hemisphere in executive functions as a cognitive substrate;(11)Assessing the impact of pragmatic failure in speech acts on pragmatic competence;(12)Investigating the patterns of PLD among learning-disabled children.

The third implication relates to the most-cited authors in PLD. The detection of such authors has the potential to enhance the appreciation of the current developments in PLD. Although a single publication is a contribution to PLD, the contribution of the most cited authors is more impactful because they have a deeper understanding of the area or have explored themes that are central to the field of PLD. Among these authors were Mentis and Prutting [129], who provided a reliable multidimensional topic analysis, whereas Solberg, Mosser, and McDonald [175] discussed measurement systems. Blum-Kulka and Snow [122] focused on developing children’s autonomy in telling stories. Taguchi [91] comprehensively dealt with teaching pragmatics. Matthews, Biney, and Abbot-Smith [51] analysed the individual differences in children’s pragmatic ability, while Olshtain and Kupferberg [145] were concerned with the professional knowledge of foreign language teachers and how it was reflected in discourse.

The fourth implication pertains to the most-cited PLD publications. Again, recognising these publications is essential because they contain topics and themes in PLD that have drawn the interest of the vast majority of researchers in this subject. Consequently, they are regarded as crucial contributions to leading PLD research. They include topics such as the effect of rejoinders in production questionnaires [176], grounds for instruction in pragmatics [177], pragmatic competence [86], minimisation and conversational inference [117], grammatical errors in language impairment [89], cross-cultural pragmatics [158], interlanguage pragmatics [88], pragmatics and language teaching [113], politeness [116], children’s contributions to dinner talk [118], teaching pragmatics [91], and discourse in the marketplace [178].

The final implication concerns the identification of authors who have the potential to be often cited by other authors. Again, this may be attributable to the fact that their research contains ideas and themes that are currently being disputed and will continue to be investigated in the future. Among the most cited items include the effect of rejoinders in production questionnaires [176], grounds for instruction in pragmatics [177], children’s autonomy in telling stories [122], research in memory [179], the principles for constructing polite speeches [116], and children’s contributions to dinner talk [118].

## 5. Practical Implications

In order for scientometric studies to be meaningful, researchers must be careful to interpret the findings closely [180], regardless of whether the use of this research method has become more popular in recent years [181,182]. It is recommended that data be retrieved from multiple sources rather than a single database, unless this is well justified (e.g., in this study we used the Scopus, WOS, and Lens databases). We believe that the next step should be to use different tools for the analysis in order to allow for the inclusion of various scientometric indicators (for example, we used both CiteSpace and VOSviewer in this study).

## 6. Theoretical Implications

At least two theoretical implications can be drawn from this study. First, we presented bibliometric and scientometric indicators highlighting that a great deal of research in PLD takes place in clinical settings. Research should be directed towards school settings and home settings in order to observe PLD in typical language development contexts and compare this evidence with that obtained in clinical settings. Second, most existing evidence focuses on children, adolescents, and adults in either first language or foreign language contexts. Studies on how new-borns and infants acquire pragmatic competence and begin to use communicative competence are limited. Although conducting research on PLD in new-borns and infants may be challenging or unmeasurable, no concrete evidence supports the immeasurability of PLD during the first years of life.

## 7. Limitations and Future Directions

There are at least two potential limitations to the results of this study. The first limitation concerns the analysis of the evidence for the most researched topics in PLD. While it was possible to group the collected data into several clusters, a detailed examination of these clusters was beyond the scope of this study. As a next step, it would be beneficial to examine these clusters in greater detail in order to identify convergences and divergences between them and thus better document the study of PLD by clinicians and researchers in clinical and educational settings. The second potential limitation is that a scientometric analysis of PLD was not able to identify the most commonly used research methodologies. Following this, a scoping review should be conducted to evaluate existing evidence on PLD based on the methods and measures used to collect and analyse data.

## 8. Conclusions

The purpose of this study was to examine the size and trends of knowledge production in PLD research. We examined 6455 PLD documents collected from Scopus, WOS, and Lens between 1950 and 2022 using CiteSpace and VOSviewer for scientometric analysis. According to our analysis, there are three major patterns in the study of PLD. Firstly, the leading research institutions and researchers in PLD seem to be centred in the United States, the United Kingdom, China, and Iran. It was beyond the scope of this study to determine whether the intensity of production and contribution to PLD from these regions can be attributed to funding, as well as individual interest in this field. Second, the most common topics examined in PLD were grouped into 12 patterns, as listed in the Discussion section. Third, taking into account the research clusters identified, it is crucial to conduct research on early PLD that considers both linguistic and non-linguistic aspects of language.

## Figures and Tables

**Figure 1 children-09-01407-f001:**
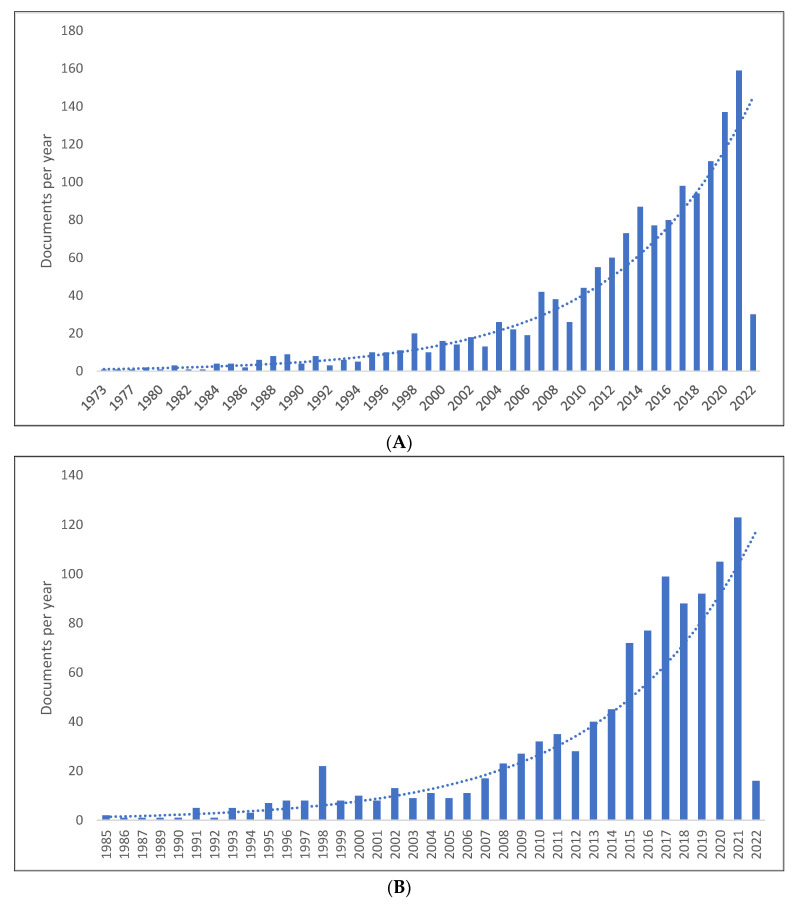
Pragmatic language development knowledge production size by year: (**A**) Scopus; (**B**) WOS; (**C**) Lens.

**Figure 2 children-09-01407-f002:**
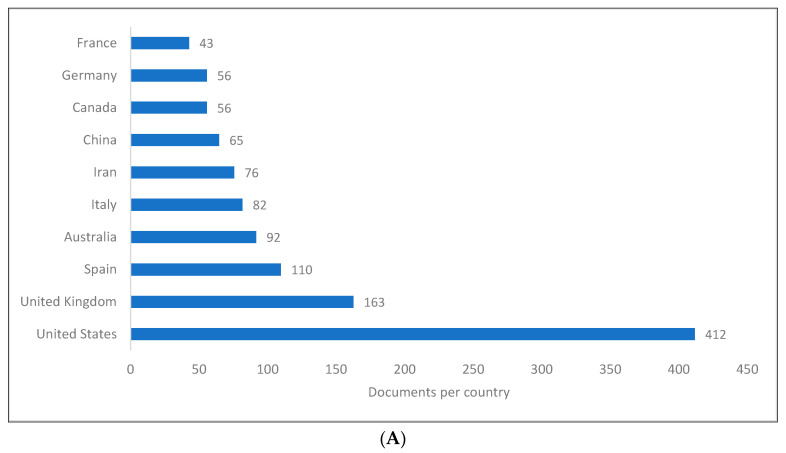
Pragmatic language development knowledge production size by country: (**A**) Scopus; (**B**) WOS; (**C**) Lens.

**Figure 3 children-09-01407-f003:**
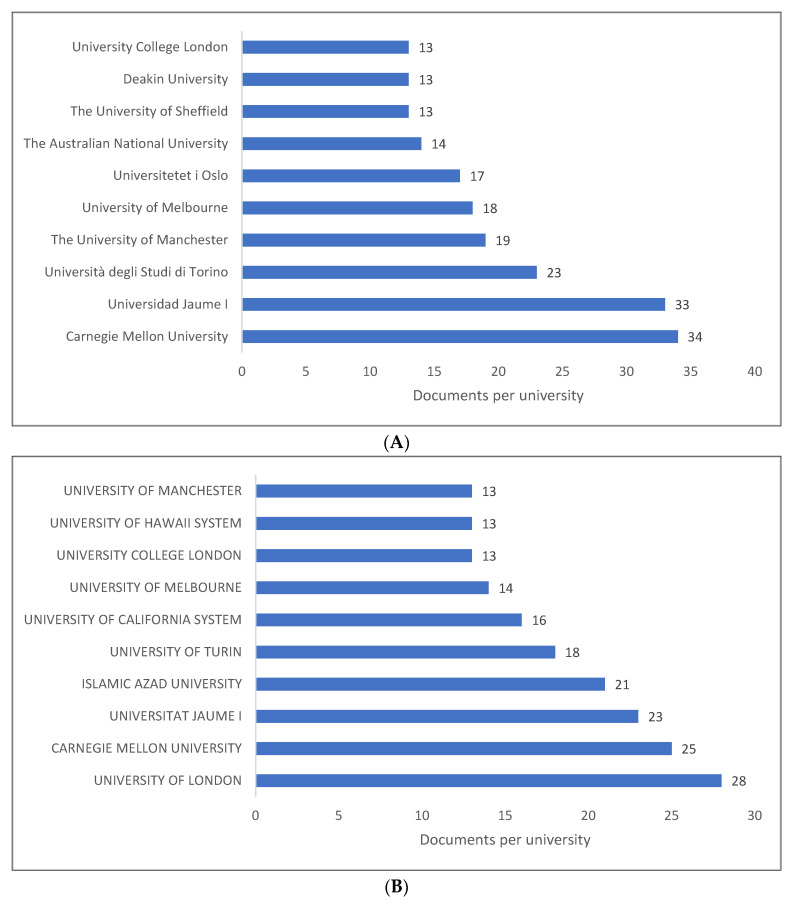
Pragmatic language development knowledge production size by university: (**A**) Scopus; (**B**) WOS; (**C**) Lens.

**Figure 4 children-09-01407-f004:**
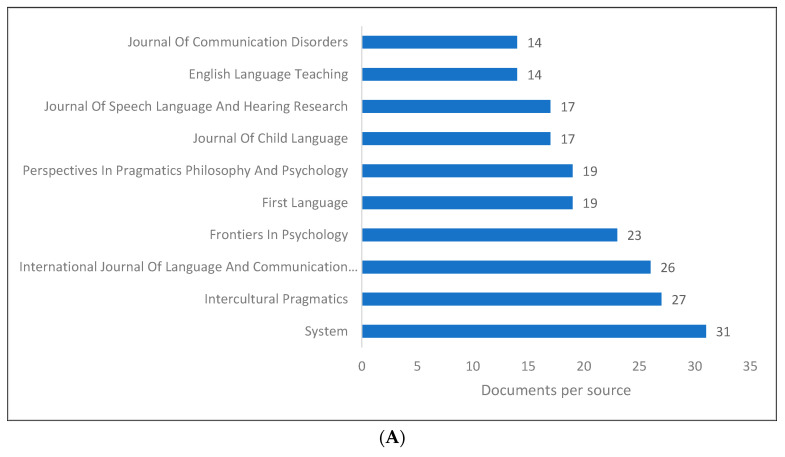
Pragmatic language development knowledge production size by journal: (**A**) Scopus; (**B**) WOS; (**C**) Lens; (**D**) Lens.

**Figure 5 children-09-01407-f005:**
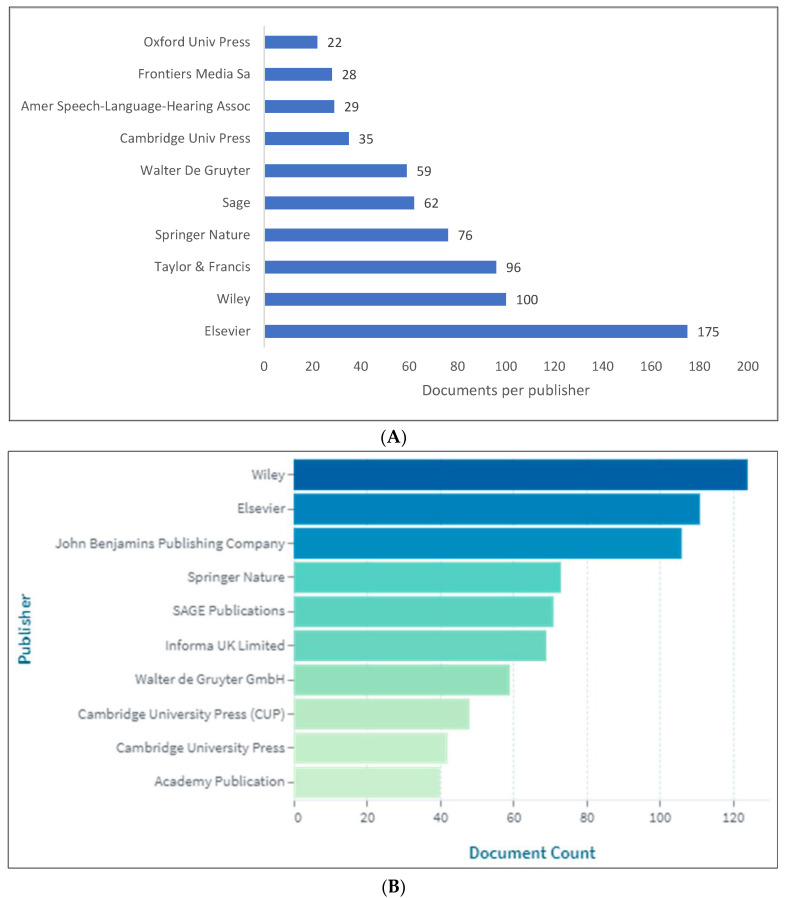
Pragmatic language development knowledge production size by publisher: (**A**) WOS; (**B**) Lens.

**Figure 6 children-09-01407-f006:**
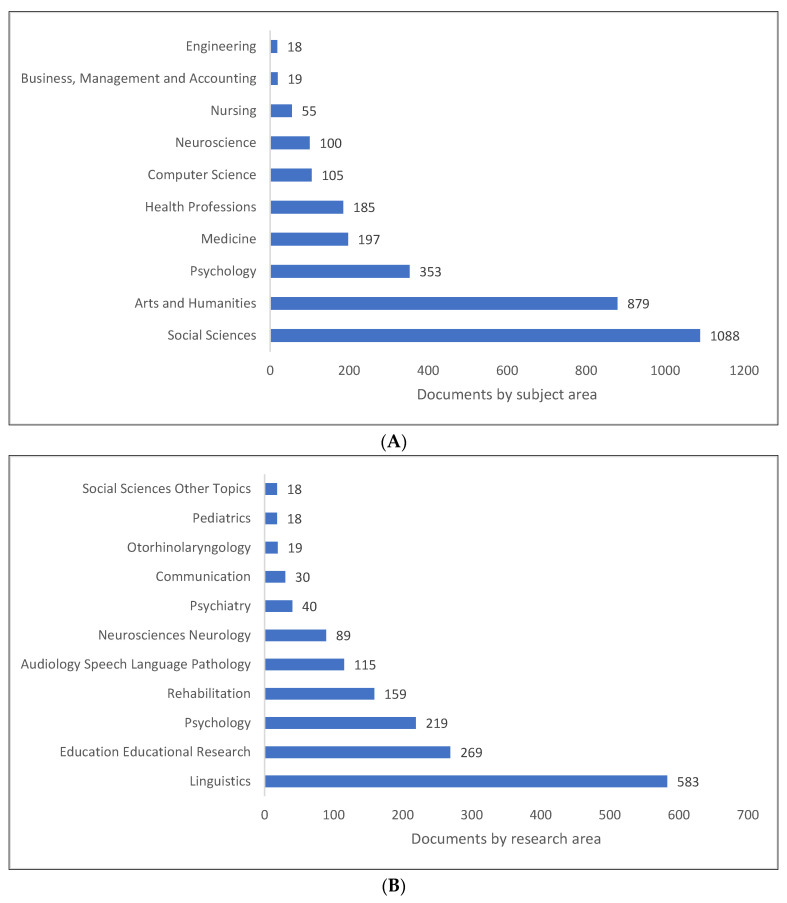
Pragmatic language development knowledge production size by research area: (**A**) Scopus; (**B**) WOS; (**C**) Lens.

**Figure 7 children-09-01407-f007:**
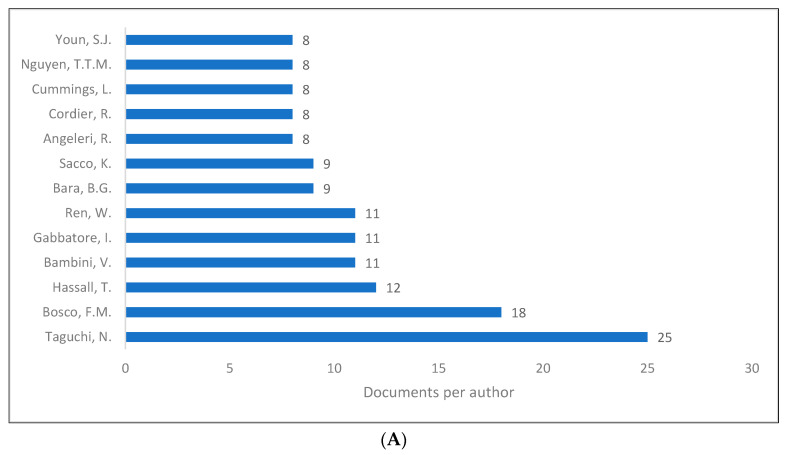
Pragmatic language development knowledge production size by author: (**A**) Scopus; (**B**) WOS; (**C**) Lens.

**Figure 8 children-09-01407-f008:**
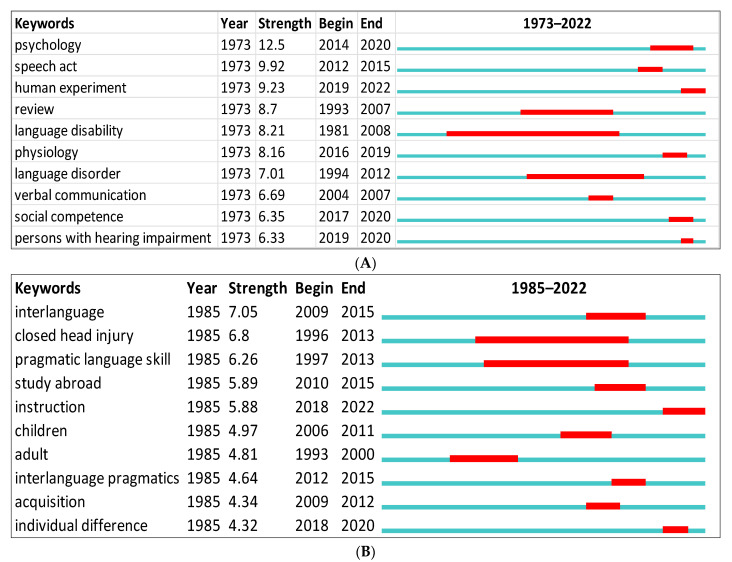
Top 10 keywords with the strongest citation bursts: (**A**) Scopus; (**B**) WOS.

**Figure 9 children-09-01407-f009:**
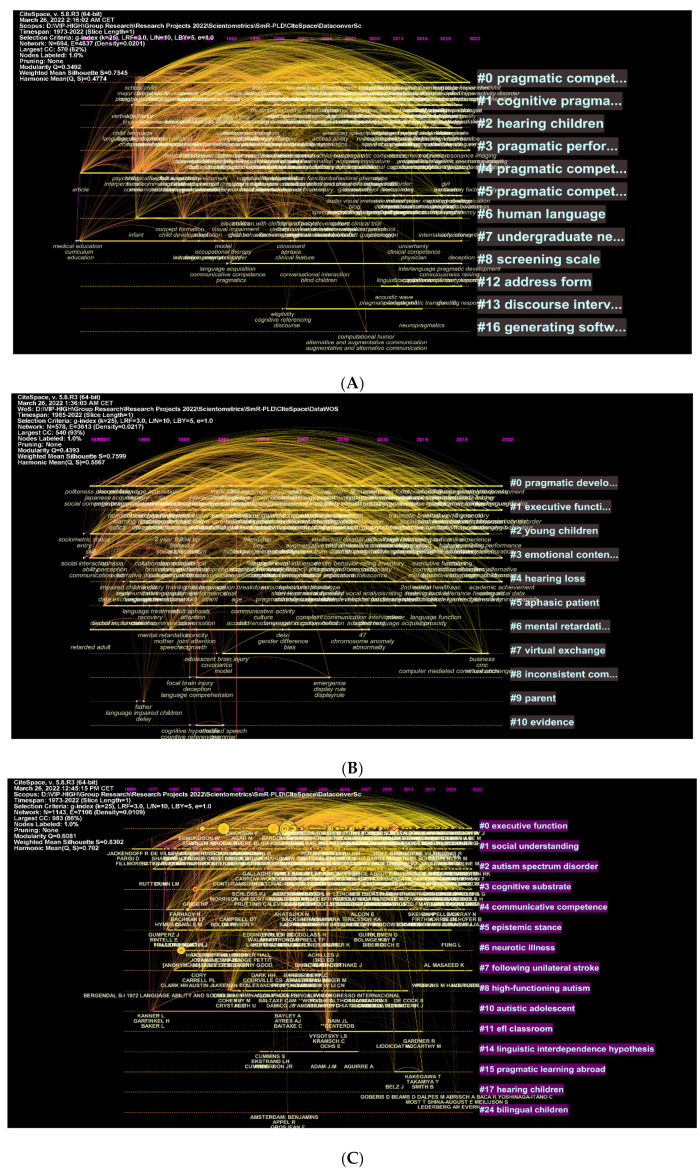
Timeline of top keywords, cited authors, and clusters: (**A**) Scopus; (**B**) WOS; (**C**) Scopus—cited authors; (**D**) WOS.

**Figure 10 children-09-01407-f010:**
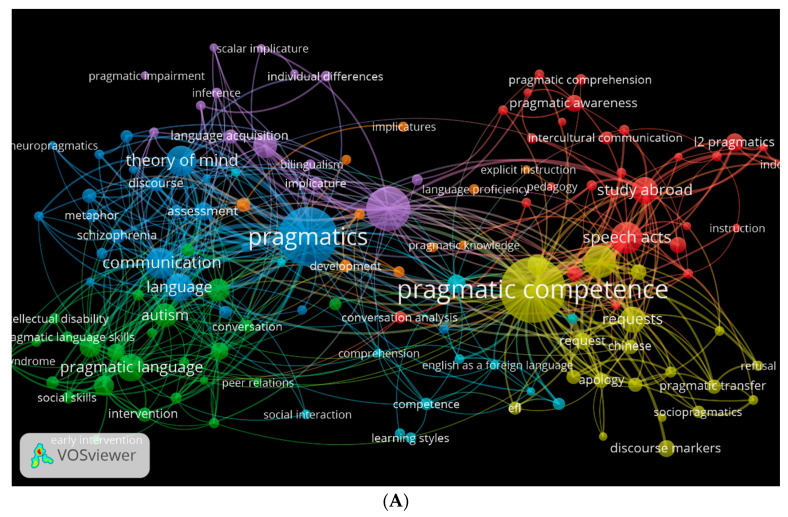
Network visualisation of keyword co-occurrence by author: (**A**) Scopus; (**B**) WOS; (**C**) Lens.

**Figure 11 children-09-01407-f011:**
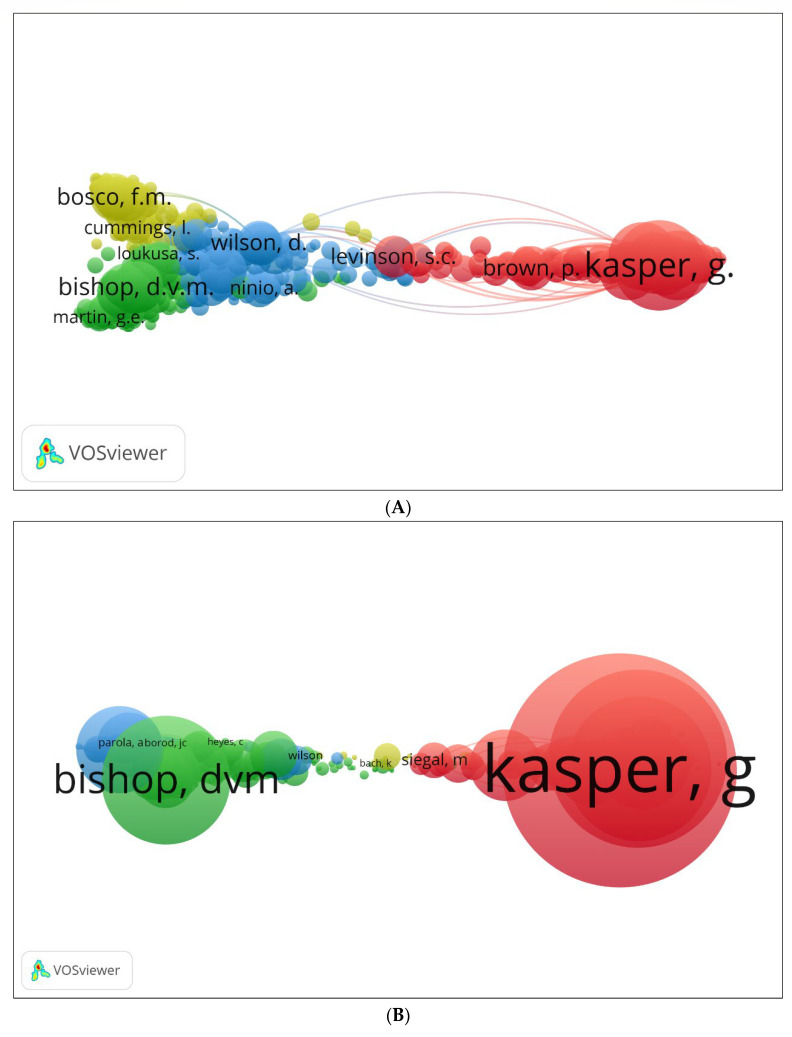
Network visualisation of author (co)-citation: (**A**) Scopus; (**B**) WOS; (**C**) Lens—citation.

**Figure 12 children-09-01407-f012:**
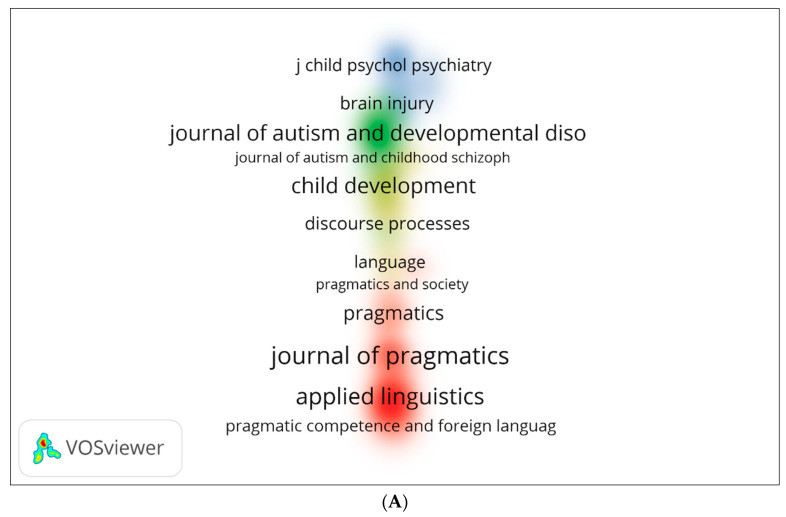
Density visualisation of (co)-citation by source: (**A**) Scopus; (**B**) WOS; (**C**) Lens—citation.

**Figure 13 children-09-01407-f013:**
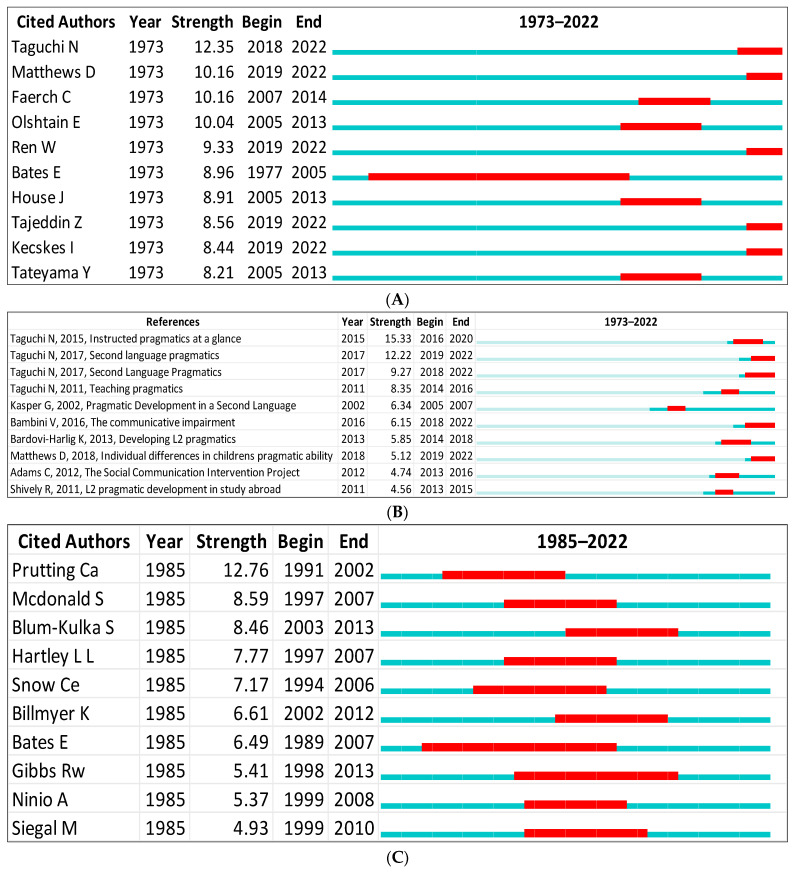
Top 10 cited authors and references with the strongest citation bursts: (**A**) Scopus [16,67,86,111,128,140,142,144,145,146]; (**B**) Scopus [51,91,147,148,149,150,151,152,153]; (**C**) WOS [8,50,92,118,129,131,132,136,138,139]; (**D**) WOS [147,148,150,151,153,154,155,156,157].

**Table 1 children-09-01407-t001:** Major communication skills of pragmatic competence (adapted from ASHA, as cited in [63] (p. 12)).

Using Language	Changing Language	Following Rules
Greetings (e.g., Hello. Goodbye. How are you?)	Used according to the needs of a listener or situation	Used for conversations and storytelling
Informing (e.g., I am leaving.)	Talking differently to a baby than to an adult	Taking turns in conversation
Demanding (e.g., Pick up the toy.)	Giving background information to an unfamiliar listener	Introducing topics of conversation
Promising (e.g., I am going to the playground.)	Speaking differently in a classroom than on a playground	Staying on topic
Requesting (e.g., Do you want to go along?)		Rephrasing when misunderstood
		Using verbal and non-verbal signals
		How close to stand to someone when speaking
		Using facial expressions
		Using eye contact

**Table 2 children-09-01407-t002:** Journals publishing research in pragmatic language development.

Source Journal	Host Country	Publisher	Span	Volumes	Web Address	Scope of the Journal
*Children*	Switzerland	MDPI AG	2020–2021	9	https://www.mdpi.com/journal/children, accessed on 10 July 2022	Sharing clinical, epidemiological, and translational science relevant to children’s health.
*Children and Youth Services Review*	United Kingdom	Elsevier Ltd.	1979–2021	18	https://www.sciencedirect.com/journal/children-and-youth-services-review, accessed on 10 July 2022	Focusing on disadvantaged or otherwise at-risk children, youth, families, and the systems supporting them. Research relevant to policies, interventions, programs, and services that enhance well-being is conducted in this forum.
*Child Development*	United States	Wiley-Blackwell	1945–1948, 1950–2021	93	https://srcd.onlinelibrary.wiley.com/journal/, accessed on 10 July 2022	Providing the latest research to researchers and theorists as well as child psychiatrists, clinical psychologists, psychiatric social workers, early childhood education specialists, school psychologists, special education teachers, and other researchers.
*Journal of* *Child Language*	United Kingdom	Cambridge University Press	1974–2021	49	https://www.cambridge.org/core/journals/journal-of-child-language, accessed on 10 July 2022	Taking a comprehensive look at children’s language behavior, the principles underlying it, and the theories that explain it.
*Language Learning and Development*	United Kingdom	Taylor and Francis Ltd.	2010–2021	18	https://www.tandfonline.com/loi/hlld20, accessed on 10 July 2022	Providing an opportunity for interaction among a wide community of scholars and practitioners interested in language acquisition.
*International Journal of Language and Communication Disorders*	United States	Wiley-Blackwell	1966–2021	57	International Journal of Language & Communication Disorders-Wiley Online Library	Speech, language, and communication disorders, as well as speech and language therapy, are all welcome.
*Journal of Speech, Language, and Hearing Research*	United States	American Speech–Language–Hearing Association (ASHA)	1996–2021	65	https://pubs.asha.org/journal/jslhr, accessed on 10 July 2022	Publishing peer-reviewed research and other scientific articles on speech, language, hearing, cognition, oral motor skills, and swallowing.
*Infants and Young Children*	United States	Lippincott Williams and Wilkins Ltd.	1988–2021	35	https://journals.lww.com/iycjournal/pages/default.aspx, accessed on 10 July 2022	Providing support to children with disabilities and their families, ages birth to five.
*Autism and Developmental Language Impairments*	United Kingdom	SAGE Publications Ltd.	2018–2021	7	https://journals.sagepub.com/home/dli, accessed on 10 July 2022	Contributing to the shaping of research in developmental communication disorders.
*Studies in Pragmatics*	Netherlands	Brill Academic Publishers	2009–2010, 2012, 2014–2017	21	https://brill.com/view/serial/SIP, accessed on 10 July 2022	Providing high-quality theoretical, analytical, and applied pragmatic studies in a widely read, respected international forum.

**Table 3 children-09-01407-t003:** Bibliometric and scientometric indicators to measure pragmatic language development.

Element	Definition/Specification/Retrieved Data	Database/Software		
Indicator		Scopus	WOS	Lens
Bibliometric				
Year	Production size by year	√	√	√
Country	Top countries publishing in the field	√	√	√
University	Top universities, research centres, etc.	√	√	√
Source	Top journals, book series, etc.	√	√	√
Publisher	Top publishers	X	√	√
Subject area	Top fields associated with the field	√	√	√
Author	Top authors publishing in the field	√	√	√
Citation	Top cited documents	√	√	√
Scientometric		CiteSpace	VOSviewer	
Betweenness centrality	Achieved when located on a path between two nodes [81].	√	X	
Burst detection	Determines the frequency of a certain event in a certain period (e.g., the frequent citation of a certain reference during a period of time) [82].	√	X	
Co-citation	When two references are cited by a third reference [83]. CiteSpace provides a document co-citation network for references and an author co-citation network for authors. In VOSviewer, co-citation is defined as “the relatedness of items is determined based on the number of times they are cited together” [80] (p. 5). Units of analysis include cited authors, references, or sources.	√	√	
Silhouette	Used in cluster analysis to measure the consistency of each cluster with its related nodes [79].	√	X	
Sigma	To measure the strength of a node in terms of betweenness centrality and citation burst [79].	√	X	
Clusters	“We can probably eyeball the visualized network and identify some prominent groupings” [79] (p. 23).	√	√	
Citation	“The relatedness of items is determined based on the number of times they cite each other” [80] (p. 5). Units of analysis include documents, sources, authors, organisations, or countries.	√	√	
Keywords	CiteSpace provides co-occurring author keywords and keywords plus. In VOSviewer, for co-occurrence analysis: “the relatedness of items is determined based on the number of documents in which they occur together” [80] (p. 5). Units of analysis include author keywords, all keywords, or keywords plus.	√	√	

**Table 4 children-09-01407-t004:** Search strings for retrieving data on pragmatic language development.

**Scopus** (TITLE-ABS-KEY (“pragmatic language development”) OR TITLE-ABS-KEY (“pragmatic development”) OR TITLE-ABS-KEY (“pragmatic language skills”) OR TITLE-ABS-KEY (“pragmatic skills”) OR TITLE-ABS-KEY (“pragmatic language competence”) OR TITLE-ABS-KEY (“pragmatic competence”) OR TITLE-ABS-KEY (“pragmatic language acquisition”) OR TITLE-ABS-KEY (“pragmatic acquisition”) OR TITLE-ABS-KEY (“pragmatic language learning”) OR TITLE-ABS-KEY (“pragmatic learning”) OR TITLE-ABS-KEY (“pragmatic language performance”) OR TITLE-ABS-KEY (“pragmatic performance”)) AND (EXCLUDE (EXACTSRCTITLE, “Journal Of Hazardous Materials”)) AND (LIMIT-TO (DOCTYPE, “ar”) OR LIMIT-TO (DOCTYPE, “ch”) OR LIMIT-TO (DOCTYPE, “re”) OR LIMIT-TO (DOCTYPE, “bk”)) Friday, 25 March 2022, 1470 document results, 1973–2022.
**WOS** “pragmatic language development” (Topic) or “pragmatic language skills” (Topic) or “pragmatic language competence” (Topic) or “pragmatic language acquisition” (Topic) or “pragmatic language learning” (Topic) or “pragmatic language performance” (Topic) or “pragmatic development” (Topic) or “pragmatic competence” (Topic) or “pragmatic acquisition” (Topic) or “pragmatic learning” (Topic) or “pragmatic performance” (Topic) or “pragmatic skills” (Topic) and JOURNAL OF HAZARDOUS MATERIALS (Exclude—Publication Titles) and Articles or Book Chapters or Review Articles or Early Access or Books (Document Types) Friday, 25 March 2022, 1063 results, 1985–2022.
**Lens** (Title: (AND (“pragmatic language development” AND)) OR (Abstract: (AND (“pragmatic language development” AND)) OR (Keyword: (AND (“pragmatic language development” AND)) OR Field of Study: (AND (“pragmatic language development” AND))))) OR ((Title: (AND (“pragmatic development” AND)) OR (Abstract: (AND (“pragmatic development” AND)) OR (Keyword: (AND (“pragmatic development” AND)) OR Field of Study: (AND (“pragmatic development” AND))))) OR ((Title: (AND (“pragmatic language skills” AND)) OR (Abstract: (AND (“pragmatic language skills” AND)) OR (Keyword: (AND (“pragmatic language skills” AND)) OR Field of Study: (AND (“pragmatic language skills” AND))))) OR ((Title: (AND (“pragmatic skills” AND)) OR (Abstract: (AND (“pragmatic skills” AND)) OR (Keyword: (AND (“pragmatic skills” AND)) OR Field of Study: (AND (“pragmatic skills” AND))))) OR ((Title: (AND (“pragmatic language competence” AND)) OR (Abstract: (AND (“pragmatic language competence” AND)) OR (Keyword: (AND (“pragmatic language competence” AND)) OR Field of Study: (AND (“pragmatic language competence” AND))))) OR ((Title: (AND (“pragmatic competence” AND)) OR (Abstract: (AND (“pragmatic competence” AND)) OR (Keyword: (AND (“pragmatic competence” AND)) OR Field of Study: (AND (“pragmatic competence” AND))))) OR ((Title: (AND (“pragmatic language acquisition” AND)) OR (Abstract: (AND (“pragmatic language acquisition” AND)) OR (Keyword: (AND (“pragmatic language acquisition” AND)) OR Field of Study: (AND (“pragmatic language acquisition” AND))))) OR ((Title: (AND (“pragmatic acquisition” AND)) OR (Abstract: (AND (“pragmatic acquisition” AND)) OR (Keyword: (AND (“pragmatic acquisition” AND)) OR Field of Study: (AND (“pragmatic acquisition” AND))))) OR ((Title: (AND (“pragmatic language learning” AND)) OR (Abstract: (AND (“pragmatic language learning” AND)) OR (Keyword: (AND (“pragmatic language learning” AND)) OR Field of Study: (AND (“pragmatic language learning” AND))))) OR ((Title: (AND (“pragmatic learning” AND)) OR (Abstract: (AND (“pragmatic learning” AND)) OR (Keyword: (AND (“pragmatic learning” AND)) OR Field of Study: (AND (“pragmatic learning” AND))))) OR ((Title: (AND (“pragmatic language performance” AND)) OR (Abstract: (AND (“pragmatic language performance” AND)) OR (Keyword: (AND (“pragmatic language performance” AND)) OR Field of Study: (AND (“pragmatic language performance” AND))))) OR (Title: (AND (“pragmatic performance” AND)) OR (Abstract: (AND (“pragmatic performance” AND)) OR (Keyword: (AND (“pragmatic performance” AND)) OR Field of Study: (AND (“pragmatic performance” AND))))))))))))))) Filters: Stemming = Disabled Publication Type = (journal article, unknown, book chapter, dissertation, book, preprint) Author Display Name = (excl John Forester) Friday, 25 March 2022, Scholarly Works (3922), 1950–2022.

**Table 5 children-09-01407-t005:** Top cited documents in pragmatic language development from Scopus, WOS, and Lens.

No.	Source Title	Citation	Citations by Database		
			Scopus	WOS	Lens
1	A new clinical tool for assessing social perception after traumatic brain injury	[92]	X	394	X
2	Accessing the unsaid: The role of scalar alternatives in children’s pragmatic inference	[93]	X	139	X
3	Acquisition in interlanguage pragmatics	[94]	X	X	314
4	An introduction to Japanese linguistics	[95]	X	X	322
5	Developmental issues in interlanguage pragmatics	[88]	212	X	391
6	Do language learners recognize pragmatic violations? Pragmatic versus grammatical awareness in instructed L2 learning	[96]	254	200	483
7	Exploring the interlanguage of interlanguage pragmatics: A research agenda for acquisitional pragmatics	[97]	199	154	X
8	Impairments in social cognition following severe traumatic brain injury	[98]	X	126	X
9	Language development: an introduction	[99]	X	X	517
10	Learning considered within a cultural context—Confucian and Socratic approaches	[100]	353	296	421
11	Narrative as a tool for the assessment of linguistic and pragmatic impairments	[101]	215	X	X
12	Narrative skills of children with communication impairments	[102]	311	273	412
13	Peer feedback on language form in telecollaboration	[103]	X	126	X
14	Pragmatic development in a second language	[104]	X	X	611
15	Pragmatics in language teaching	[105]	X	X	590
16	The comment clause in English: Syntactic origins and pragmatic development	[106]	206	X	X
17	The social bases of language acquisition	[107]	208	X	410
18	Transfer in bilingual development—the linguistic interdependence hypothesis revisited	[108]	X	150	X
19	Trolling in asynchronous computer-mediated communication: From user discussions to academic definitions	[109]	310	X	X
20	What we’re teaching teachers: An analysis of multicultural teacher education coursework syllabi	[110]	186	151	X

**Table 6 children-09-01407-t006:** Summary of the largest clusters of pragmatic language development.

Cluster ID	Size	Silhouette	Label (LSI)	Label (LLR)	Label (MI)	Average Year
Scopus						
0	236	0.737	pragmatic development	executive function (781.1, 1.0 × 10^−4^)	social behaviour (3.89)	2009
1	164	0.742	pragmatic development	social understanding (520.14, 1.0 × 10^−4^)	social behaviour (0.75)	1994
2	135	0.833	autism spectrum disorder	autism spectrum disorder (1378.46, 1.0 × 10^−4^)	social behaviour (1.18)	2007
3	120	0.862	executive function	executive function (667.96, 1.0 × 10^−4^)	academic setting (1.35)	2010
4	59	0.904	pragmatic competence	communicative competence (527.75, 1.0 × 10^−4^)	social behaviour (0.46)	2002
5	55	0.879	pragmatic language skill	epistemic stance (252.15, 1.0 × 10^−4^)	aphasic patient (0.19)	1996
WOS						
0	214	0.737	pragmatic development	foreign language (729.41, 1.0 × 10^−4^)	behavioural problem (3.33)	2009
1	147	0.853	autism spectrum disorder	autism spectrum disorder (938.03, 1.0 × 10^−4^)	behavioural problem (2.13)	2007
2	110	0.942	traumatic brain injury	closed head injury (358.23, 1.0 × 10^−4^)	behavioural problem (0.37)	1998
3	77	0.896	executive function	cognitive substrate (339.59, 1.0 × 10^−4^)	right hemisphere (0.7)	2010
4	67	0.905	pragmatic competence	speech act (547.33, 1.0 × 10^−4^)	pragmatic failure (0.76)	2000
5	62	0.955	learning-disabled children	learning-disabled children (93.91, 1.0 × 10^−4^)	pragmatic development (0.03)	1993

**Table 7 children-09-01407-t007:** Top citations counts for pragmatic language development.

WoS			Scopus		
Citation	Reference	Cluster ID	Citation	Reference	Cluster ID
258	Kasper [114]	0	380	Kasper [88]	0
187	Bardovi-Harlig [115]	0	300	Bardovi-Harlig [113]	0
162	Taguchi [86]	0	254	Brown [116]	0
153	Levinson [117]	0	239	Blum-Kulka [118]	0
131	Bishop [89]	1	220	Taguchi [91]	0
113	House [119]	0	181	[Anonymous], 1981	6
113	Thomas [120]	0	178	Thomas [121]	0
111	Blum-Kulka [122]	0	145	Ellis [123]	0
109	Barron [124]	0	145	Grice [125]	3
102	Ellis [126]	0	145	House [127]	0

**Table 8 children-09-01407-t008:** Top bursts detected in pragmatic language development.

WoS			Scopus		
Burst	Reference	Cluster ID	Burst	Reference	Cluster ID
12.76	Prutting [129]	1	12.35	Taguchi [91]	0
8.59	Mcdonald [130]	2	10.16	Faerch [128]	0
8.46	Blum-Kulka [122]	0	10.16	Matthews [131]	3
7.77	Hartley [132]	2	10.04	Olshtain [133]	0
7.17	Snow [134]	7	9.33	Ren [135]	0
6.61	Billmyer [136]	0	8.96	Bates [137]	1
6.49	Bates [138]	1	8.91	House [127]	0
5.41	Gibbs [139]	3	8.56	Tajeddin [140]	0
5.37	Ninio [141]	1	8.44	Kecskes [142]	0
4.93	Siegal [143]	6	8.21	Tateyama [144]	0

**Table 9 children-09-01407-t009:** Betweenness centrality for top 10 authors in pragmatic language development.

WoS			Scopus		
Centrality	Reference	Cluster ID	Centrality	Reference	Cluster ID
97	Kasper [114]	0	148	Bates [137]	1
96	Bardovi-Harlig [115]	0	120	Brown [116]	0
85	Blum-Kulka [122]	0	103	Blum-Kulka [118]	0
78	Blum-Kulka [158]	0	102	Bardovi-Harlig [113]	0
76	Takahashi [159]	0	100	Kasper [88]	0
73	Felix-Brasdefer [160]	0	91	Ellis [123]	0
72	Bishop [89]	1	90	Schmidt [161]	0
70	Taguchi [162]	0	81	Achiba [163]	0
70	Blum-Kulka [164]	0	81	Takahashi [165]	0
69	House [146]	0	81	Rose [149]	0

**Table 10 children-09-01407-t010:** Sigma metrics for top 10 authors in pragmatic language development.

WoS			Scopus		
Sigma	Reference	Cluster ID	Sigma	Reference	Cluster ID
0	Kasper [114]	0	0	Bates [137]	1
0	Bardovi-Harlig [115]	0	0	Brown [116]	0
0	Blum-Kulka [122]	0	0	Blum-Kulka [118]	0
0	Blum-Kulka [158]	0	0	Bardovi-Harlig [113]	0
0	Takahashi [159]	0	0	Kasper [88]	0
0	Felix-Brasdefer [160]	0	0	Ellis [123]	0
0	Bishop [89]	1	0	Schmidt [161]	0
0	Taguchi [162]	0	0	Achiba [163]	0
0	Blum-Kulka [164]	0	0	Takahashi [165]	0
0	House [146]	0	0	Rose [149]	0

## Data Availability

The data presented in this study are available on request from the first author.

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
