# Peer review of "Pragmatic Language Development: Analysis of Mapping Knowledge Domains on How Infants and Children Become Pragmatically Competent"

_children, 2022, doi:10.3390/children9091407_

Round 1

Reviewer 1 Report

I already reviewed an earlier version of this manuscript and rejected it from publication as having very low merit. I am quite surprised to see it resubmitted again with very little change. The manuscript does very little to contribute to the new significant knowledge regarding the PDL. Out of three questions the manuscript asks: i.e. “1) Based on the size of production and the number of (co)citations, which regions, institutions, authors, and documents are considered to be the most influential? 2) Which topics and themes are most frequently explored in PLD? 3) Which research trends are emerging in PLD? “ Only the least interesting – i.e. Question 1 is answered at length. However, why is it so important to the readers to know the locations that generate PLD knowledge is not made clear. The most interesting questions – topics and themes and emerging trends are not coming through and lost in the unnecessary details and extraneous points. 

Reviewer 2 Report

The article is proficient enough, provide a lot of evidence and illustrations. It is well-structured and provisions are supported by evidence of any kind. References relate to the main content of the article and don't look abundant. 

Reviewer 3 Report

Dear Authors, the study is indeed extensive and comprehensive, with a detailed and pertinent overview of the methods used in discursive and visual representations (e.g. network visualisations). Another strong point concerns the integration of pragmatics (and PLD) with other fields that transpired from your data analysis and that would ultimately lead to multiple areas of convergence to be explored further, as you clearly pointed out in your paper. I also consider the research methods and methodology consistent and fully detailed. Furthermore, the mapping of knowledge domains might be of immense relevance for scientists in the field

As far as the areas of improvement, I would suggest the following points to be considered by the authors towards their final publication of the paper:

1. Page setup and format- they seem disorderly and incosistent to a certain extent. 

2. Introduction- it would benefit from a better organisation of ideas, as they seem repetitive (paragraph 1, for instance).

3. Title p. 13- Measures?- Wouldn't 'measurements' be a better choice?

4. p. 14. - I would recommend improving the clarity of paragraph 1 (e.g. the second reason)

5. overall consistency of spelling (see: cooccurrence vs. co-occurrence)

6. on p. 20- it is not particularly clear to me how the selection was performed for the contributing university/institute (on 1st author basis/ corresponding author basis- especially for joint research teams)

7- on p. 29- production of PLD- paragraph 1 could benefit from motivation and clarity improvement.

8. The overall purpose of the study is to provide a quantitative overview of the literature, which it achieves. However, I would like to see better integration and discussion of the themes and topics associated with this field (as they are only briefly mentioned in the abstract and conclusion, but even less briefly in the corpus of the article and discussions) in the sense that a more detailed guideline onto these themes and topics might serve the purpose of this article and future research in the field even better.

9. It is relevant to point out that the discussions and conclusions parts could be more thorough (as for example, more information on the practical implications of the research for this field).

Round 2

Reviewer 1 Report

The revisions improved the manuscript. I recommend it for publishing with Children journal.